# *Sambucus nigra* L. (fam. Viburnaceae) in Sicily: Distribution, Ecology, Traditional Use and Therapeutic Properties

**DOI:** 10.3390/plants12193457

**Published:** 2023-09-30

**Authors:** Giovanna Sala, Salvatore Pasta, Antonella Maggio, Tommaso La Mantia

**Affiliations:** 1Department of Agricultural, Food and Forest Sciences, University of Palermo, Viale delle Scienze, Ed.4, 90128 Palermo, Italy; tommaso.lamantia@unipa.it; 2Institute of Biosciences and BioResources, Italian National Research Council, via Ugo La Malfa 153, 90146 Palermo, Italy; salvatore.pasta@ibbr.cnr.it; 3Department of Biological, Chemical and Pharmaceutical Sciences and Technologies (STEBICEF), Section of Chemistry, University of Palermo, 90128 Palermo, Italy; antonella.maggio@unipa.it; 4NBFC, National Biodiversity Future Center, 90133 Palermo, Italy

**Keywords:** ethnobotany, place names, plant names, riparian vegetation, shrublands, traditional medicine

## Abstract

*Sambucus nigra*, the elderberry, has long been used for its medicinal properties in treating numerous diseases. Based on this traditional knowledge, its different pharmacological activities have been the focus of active research. All parts of the tree have long been used in traditional medicine, that is, the bark, the leaves, the flowers and the fruit. This study, carried out in Sicily (Italy), concerns the traditional uses of elder against human diseases. In order to trace the history of man’s interaction with elder on the island, multidisciplinary research was carried out, aiming at (1) presenting a comprehensive overview of elderberry’s applications and activities and (2) bridging traditional knowledge (uses and beliefs) with modern science, i.e., the most recent scientific findings in the biomedical and pharmacological fields. A rigorous literature review of scientific (and other local) reports on the elderberry tree and its application in food, health and household applications was undertaken. This article also provides a synthetic and updated picture of the ecology and distribution of *S. nigra* in Sicily. The elderberry is quite widespread in Sicily, yet its distribution is discontinuous. It prefers hedges, riparian woodlands, forest margins and clearings and is rather common along the watercourses flowing in the canyons of the Hyblaean Plateau, in the Madonie Mts. and in Enna province. Indeed, many old plants are often found near sacred places and rural houses, suggesting that in the past, it was extensively planted on purpose for its multiple uses. The complementary data obtained from multidisciplinary research confirm the usefulness of this approach in building a comprehensive and correct picture of the distribution of the most common woody species, for which the available knowledge is often fragmentary and imprecise.

## 1. Introduction

In recent years, the manyfold usages of many species used in the traditional pharmacopeia have been rediscovered. In this framework, international organizations such as the World Health Organization are encouraging this trend and promoting research into the traditional use of medicinal plants, the identification of their therapeutic properties, the development of standards for product purity and the activity of galenic preparations [1]. Some components present in plants also used by the food industry are arousing considerable scientific interest due to their potential positive effects on maintaining human wellbeing [2,3,4]. The consumption of natural remedies has increased significantly in recent years, especially in developed countries [2], and, consequently, the investment of pharmaceutical companies in terms of research and projects aimed at identifying plants of economic interest, as well as the industrial-scale production of active ingredients of plant origin, has increased [5,6]. One of the main goals of current research focusing on the chemistry of natural substances is the identification of biologically active plant metabolites commonly used in the human diet and which, in past centuries, were not only used as food or spices but also in popular medicine [7,8].

Among the neglected plants that are part of the European pharmacopeia, elderberry (*Sambucus nigra* L., fam. Viburnaceae) has played a role of primary importance for a long time [9,10]. While several parts of this plant are still used in different parts of the European continent, at present, its use in Sicily appears rather sporadic, a fact that sharply contrasts with its widespread yet scattered occurrence on the island.

This study aims at presenting a comprehensive overview of elderberry’s applications and activities, trying to bridge traditional knowledge with modern science. More in detail, multidisciplinary research was carried out to trace the history of the relationship between man and elder on the island, looking at knowledge related to the uses and popular beliefs connected with elderberry in light of the most recent scientific research in the biomedical and pharmacological fields.

This article also provides a synthetic and updated picture of the ecology and distribution of *S. nigra* in Sicily.

### Distribution Range and Ecology of Sambucus nigra

It is difficult to trace the native distribution range of *S. nigra* because it has been widely used by man since ancient times. Today, elderberry is common throughout central Europe, from the Atlantic coasts of Ireland and the Iberian Peninsula to those of the Black Sea and eastwards in the Iranian–Turanian and Caucasian areas up to a longitude of 55° east [11,12]. Being less common in the Mediterranean area, *S. nigra* does not occur in southern Greece and Crete [13]. In more recent times, it was introduced and became naturalised in the Scandinavian and Baltic countries, reaching a latitude of 63° north along the coasts of western Norway and about 55° north in Lithuania, as well as in northwestern Africa (Morocco, Algeria and Tunisia), the Azores, the United States, East Asia, South Australia and New Zealand [14,15].

In ancient Greece, the seeds of elderberry were an ingredient of Demeter’s ritual; its names, ακτέα (aktea) or ακτέ (akté), which are probably derived from the ancient Indo-European root that gave rise to the Sanskrit “açnati” (to eat), suggest that its berries were consumed by humans even before the spread of cereals during the Neolithic period [16]. Despite the numerous archaeobotanical investigations carried out across Sicily over the last few decades, the presence of pollen from *Sambucus* sp. was recorded in only three sites, i.e., in the sediments of the lake of Pergusa, dating back to ca. 10,815 ±160 years ago [17], in the Bronze Age village (first half of the 2nd millennium BC) of San Vincenzo at Stromboli Island in the Aeolian archipelago [18] and in the late Roman and Byzantine levels (400–800 AD) of the rock shelter of Valle Inferno on the Madonie Massif [19].

Besides human introduction, the spread of elderberries might also have benefitted from bird dispersal [20,21]. In this regard, although data referring to the Sicilian territory are lacking, during bird ringing activity, operators intentionally install mist nets near fruiting elderberry trees to increase the number of catches (R. Ientile, pers. comm.).

The elder prefers open and well-lit areas on soft and humid soils rich in decomposed organic matter, from neutral to basic [20]. *S. nigra* is mainly found on riverine alluvial terraces and along forest edges and clearings. It frequently grows in hedges, along the boundaries of orchards and fields and in suburban ruderal areas (Figure 1). Due to its opportunistic pioneer behaviour, it never forms large stands, and it gradually disappears once forest communities get denser. In temperate Europe, where it is more common and certainly indigenous, *S. nigra* behaves as a fast-growing and heliophilous species, being particularly common in hedge communities and often making small pure populations in natural or artificial deciduous broad-leaved forests (e.g., oak and beech woods) ascribed to the phytosociological class *Carpino-Fagetea sylvaticae* Jacuks and Passarge 1968. In some of the most recent and prominent works on Italian and European vegetation [22,23], elderberry is featured among the species typical of mantle communities (class *Crataego-Prunetea* R. Tx. 1962), although it is also frequent in Middle-European (class *Alno glutinosae-Populetea albae* P. Fukarek and Fabijanić 1968) and Mediterranean (class *Salicetea purpureae* Moor 1958) riparian communities, as well as in the nitro-schiaphilous communities of forest clearings (class *Epilobietea angustifolii* R. Tx. and Preising ex von Rochow 1951).

## 2. Results and Discussion

The map in Figure 2 shows the spatial distribution of Sicilian vernacular names that refer to elderberry, also reported in Table 1. Many orthographic variants of the main vernacular name referring to *S. nigra*, deriving from the Latin name *sambucus*, such as “sa(m)bùc(c)-”, “sambùgh-”, “sa(m)mùgh- “, “sa(m)mùc (c)-”, “savoc(c)-”,”sa(v)ùc(c)-” and “zzammùc(c)-”, were detected. Other place names possibly referring to elderberry derive from languages of other people who lived in Sicily during last millennia. Among the vernacular names of non-Latin origin that could also refer to *S. nigra*, the Greek φλασκιά (=flaskiá) deserves to be mentioned because it may have originated the name of Torrente Flascio, the main tributary of the Simeto river, whose catchment area is located on the southern slopes of the Nebrodi Mts., while the French sureau could be the base of Suro, a place name that refers to a watershed near Gratteri on the Madonie Mts.

Figure 2 also includes all the localities listed in Table 2, Table 3 and Table 4. Table 2 shows all the regional place names probably linked to the presence of elderberries and featured on the maps of the IGMI. Both of these tables underline the frequency of this species in the hilly areas of Sicily, especially in the eastern portion of the island, while it is increasingly rare or absent at low altitudes in western Sicily and on the satellite islands.

The toponyms “Sa(b)bucina” (municipality of Caltanissetta) and “Sambucia” (municipality of Palermo) should be discarded because they probably derive from the Arabic az-zabbuğ (from where the modern Spanish “acebuche” originates), which refers to the wild olive tree *Olea europaea* L. var. *sylvestris* (Mill.) Lehr. Although there are alternative hypotheses about the etymology of the town of Sambuca di Sicilia (Agrigento province), its name seems to derive from the name of an Arab leader, Zabut, and therefore has no connection with elderberry (although small stands of *S. nigra* occur close to the nearby Lago Arancio, authors’ note).

The etymology of Sàvoca (Messina province), a small village of the Peloritani as well as that of the stream called Torrente Savoca, remains uncertain. Indeed, many other homonymous place names such as “Case Savoca” (Enna and Piazza Armerina) or “Piano Savoca” (Messina) could derive from a surname that is widespread in the provinces of Enna, Catania and Messina. In fact, two-thirds of all the Italian families named “Savoca” currently live there “https://www.cognomix.it/mappe-dei-cognomi-italiani/SAVOCA/SICILIA” (accessed on 9 January 2023).

The toponym “Sauci” is ambiguous because it could also be referred to as “sàuciu”, “sàuci” (=Salix spp.), like in the case of Portella and Monte Sàuci or Contrada Sàuci Grande and Sàuci Piccolo in San Vito Lo Capo (Trapani province). The place names characterised by the root “sambuc” and diminutive suffixes such as “-ello”, “-elli”, “-ellu”, “-(i)edd(r)i” and “-(i)edd(r)u”, found on the Nebrodi Mts. (e.g., Caronia, Cesarò, Pettineo), or “-etto”, “-ettu” and “-etti” (Nicosia) refer more probably to the “small” (i.e., annual) elder, *Sambucus ebulus* L., a rather common nitrophilous and ruderal species. The same goes for the phytonyms “savucheddu”, “saucheddu”, “sauchettu”, “saiuchieddu”, “sambuchettu” found in [24].

### 2.1. Distribution and Ecology of Elderberry on the Island

As shown in Table 3, *S. nigra* is rather widespread in Sicily, yet its distribution is discontinuous. In fact, it prefers hedges, riparian woodlands, forest margins and clearings and is by far more frequent in inland areas even at high altitudes [26]. As for the local altitudinal range of the species, in Sicily, *S. nigra* occurs between 200 and 1300 m a.s.l., preferring riparian habitats at lower altitudes and forest fringes at higher elevations. Following the Rivas-Martínez classification, it shows a very broad climatic amplitude, being able to grow under thermo-, meso- and supra-mediterranean bioclimatic conditions [27]. With no doubt, *S. nigra* behaves as a light- and nutrient-demanding pioneer woody species. Unfortunately, no reliable information on the local species’ soil requirements is available. Some authors of the early 19th century [28] stated that elderberry grew “everywhere” on Etna Mt., while 15 years ago, other authors [29] reported it as “uncommon” on the volcano. Sicilian botanists [30,31] put *S. nigra* among the characteristics of poplar-dominated riparian woodlands, referring to the order *Populetalia albae* Br.-Bl. 1962, recently ascribed to the class *Alno glutinosae–Populetea albae*. The data shown in Table 3 also point out the very high frequency of elderberry in woody hedge- (*Crataego-Prunetea*) and willow-dominated (*Salicetea purpureae*) riparian communities. Indeed, elderberries, especially older ones, are often found near convents, monasteries, hermitages and rural houses, reflecting the custom—at least up to the recent past—of cultivating this plant for multiple purposes.

**Table 3 plants-12-03457-t003:** List of the Sicilian localities where the presence of nuclei or individuals of elderberry is reported and/or has been established. PC = phytosociological reference class in accordance with that proposed in [32], namely: AG-PA = *Alno glutinosae*-*Populetea albae*; C-P = *Crataego-Prunetea*; EA = *Epilobietea angustifolii*; QI = *Quercetea ilicis*; SP = *Salicetea purpureae*; n.a. = not available.

Location	Province	Habitat	Syntaxon	Pc	Reference
SICANI MTS.					
Santo Stefano di Quisquina	Agrigento	n.a.	n.a.	n.a.	[33]
Eremo di Santa Rosalia, Santo Stefano di Quisquina	Agrigento	Forest clearing	*Pruno-Rubion ulmifolii*	C-P	S. Pasta and G. Garfì, IV.2013
Bivona	Agrigento	n.a.	n.a.	n.a.	[33]
Eremo di Sant’Adriano, Palazzo Adriano	Palermo	Forest clearing	*Pruno-Rubion ulmifolii*	QI	S. Pasta and S.D. La Mela Veca, 6.IX.2003
Gole del Torrente Frattina, Corleone	Palermo	Streamside	*Ulmo canescentis–Salicetum pedicellatae*	SP	[34]
Contrada Arancio, Sambuca di Sicilia	Agrigento	Forest clearing	*Acantho mollis–Lauretum nobilis*	QI	[35]
Monti Sicani sud-orientali	Agrigento	n.a.	n.a.	n.a.	[36]
Valle del Sosio	Agrigento	n.a.	n.a.	n.a.	G. Clementi, *com. pers*.
Monte delle Rose	Agrigento	n.a.	n.a.	n.a.	G. Clementi, *com. pers*.
IBLEI MTS.					
Strada Statale Giarratana-Ragusa, destra idrografica dell’Irminio dopo la Diga Santa Rosalia	Ragusa	Riverside	n.a.	n.a.	S. Pasta and J. Rühl, 23.IV.2006
Contrada Sant’Andrea, Buccheri	Siracusa	Forest clearing	*Hedero helicis–Lauretum nobilis*	QI	[37]
Torrente Sughereta, bacino del Fiume San Leonardo, Buccheri	Siracusa	Streamside	*Roso sempervirentis–Populetum nigrae*	AG-PA	[31]
Valle di Pietra, Buscemi	Siracusa	Forest clearing	*Hedero helicis–Lauretum nobilis*	QI	[37]
Fiume Cassibile	Siracusa	Riverside	n.a.	n.a.	[38]
Fiume Anapo	Siracusa	Riverside	*Platano orientalis–Salicetum pedicellatae*	SP	[39]
Fiume Manghisi-Cassibile	Siracusa	Streamside	*Platano orientalis–Salicetum pedicellatae*	SP	[39]
Fiume Irminio	Ragusa	Riverside	*Salicetum albo-pedicellatae*	SP	[40]
Contrada San Filippo, bacino del Fiume Irminio	Ragusa	Riverside	*Platano orientalis–Salicetum pedicellatae*	SP	[31]
Contrada Pirainito, Fiume Tellaro	Siracusa	Riverside	*Platano orientalis–Salicetum pedicellatae*	SP	[31]
Valloni San Leonardo e Santa Domenica, sponde del Fiume Irminio	Ragusa	Streamside	n.a.	n.a.	[41]
Contrada Misericordia, Fiume Irminio	Ragusa	Riverside	*Roso sempervirentis–Populetum nigrae*	AG-PA	[31]
Contrada Amerillo, Monterosso Almo	Ragusa	Streamside	*Roso sempervirentis–Populetum nigrae*	AG-PA	[31]
Contrada San Marco, Manghisi-Cassibile	Siracusa	Streamside	*Roso sempervirentis–Populetum nigrae*	AG-PA	[31]
Contrada Canalazzo, Monterosso Almo	Ragusa	Streamside	*Roso sempervirentis–Populetum nigrae*	AG-PA	[31]
Cava Cinque Porte, Manghisi-Cassibile	Siracusa	Streamside	*Roso sempervirentis–Populetum nigrae*	AG-PA	[31]
Torrente Ossena, bacino del Fiume San Leonardo, Scordia	Catania	Streamside	*Salicetum albo-pedicellatae*	SP	[31]
Torrente Sperone, bacino del Fiume San Leonardo, Scordia	Catania	Streamside	*Salicetum albo-pedicellatae*	SP	[31]
Fiume Dirillo	Ragusa	Riverside	*Roso sempervirentis–Populetum nigrae*	AG-PA	[42]
NEBRODI MTS.					
Contrada Tassita	Messina	Forest clearing	n.a.	C-P	S. Pasta and G. Garfì, VI.2013
Torrente Galati, San Basilio lungo le Gole del Catafurco	Messina	Streamside	n.a.	n.a.	[43]
Contrada Monacelle lungo il Torrente Suta; lungo il Torrente Fiumetto; lungo il torrente di Contrada Villa, Galati Mamertino	Messina	Streamside	*Petagneetum gussonei*	EA	[43]
MADONIE MTS.					
Monte Cervi, Polizzi Generosa	Palermo	Mantle community	*Lonicero xylostei–Prunetum cupanianae*	C-P	[44]
Portella di Mele	Palermo	Mantle community	*Crataegetum laciniatae*	C-P	[44]
Vallone Madonie	Palermo	Mantle community	*Crataegetum laciniatae*	C-P	[45]
Vallone San Nicola, Polizzi Generosa	Palermo	Streamside	*Ulmo canescentis–Salicetum pedicellatae*	SP	[30]
Dintorni di Petralia	Palermo	n.a.	n.a.	n.a.	[46]
Gonato, Castelbuono	Palermo	n.a.	n.a.	n.a.	Herb. Minà Palumbo [47]
Pietà, Polizzi Generosa	Palermo	n.a.	n.a.	n.a.	Herb. Minà Palumbo [47]
Barraca, Castelbuono	Palermo	n.a.	n.a.	n.a.	Herb. Minà Palumbo [47]
Bocca di Cava, Castelbuono	Palermo	n.a.	n.a.	n.a.	Strobl [47]
Monte Ferro	Palermo	n.a.	n.a.	n.a.	Strobl [47]
Isnello	Palermo	n.a.	n.a.	n.a.	Strobl [47]
Acqua del Faggio, Isnello	Palermo	n.a.	n.a.	n.a.	Strobl [47]
Scaletta del Capriolo	Palermo	n.a.	n.a.	n.a.	Tineo [47]
Crinali di Cozzo Morto e Pizzo Taverna, San Mauro Castelverde	Palermo	Forest clearing	n.a.	C-P	S. Pasta and A. Pipitone, V.2003
Contrada Sant’Andrea, Contrada Magazzeno, C. Fontana, Geraci Siculo	Palermo	n.a.	n.a.	n.a.	[48]
ETNA MT.					
Cava Grande, Sant’Alfio	Catania	Forest clearing	*Aceri obtusati–Ostryetum carpinifoliae*	QI	[49]
Presso Convento di San Nicolò, Nicolosi	Catania	n.a.	n.a.	n.a.	G. Siracusa http://www.dipbot.unict.it/herbarium/erbario.aspx (accessed 3 June 2022)
Bronte e Bosco Maletto	Catania	n.a.	n.a.	n.a.	Strobl [47]
Cavaleri	Catania	n.a.	n.a.	n.a.	Tornabene [47]
TRAPANI MTS.					
Contrada Runzi, Erice	Trapani	Mantle community	*Rubo ulmifolii–Tametum communis*	C-P	[50]
Monte Inici, Castellamare del Golfo	Trapani	Mantle comunity	*Pruno-Rubion ulmifolii*	C-P	[50]
ENNA					
Pergusa	Enna	Lake shore	n.a.	C-P	[51]
Monte Altesina	Enna	n.a.	n.a.	n.a.	[41]
Fiume Morello	Enna	Riverside	n.a.	SP	S. Pasta, A. Troia e G. Pirrera, VI.1990
Masseria Grimaldi, Contrada Santo Spirito, Enna	Enna	n.a.	n.a.	n.a.	[48]
Contrada Valle Pagana, Contrada Pietrelunghe, Portella Malamogliera, Calascibetta	Enna	n.a.	n.a.	n.a.	[48]
CALTANISSETTA					
ZSC ITA050009 “Rupe di Marianopoli”	Caltanissetta	n.a.	n.a.	C-P	[52]
PELORITANI MTS.					
Torrente Licopeti, Roccella Valdemone	Messina	Streamside	n.a.	n.a.	G. Spampinato http://www.dipbot.unict.it/herbarium/erbario.aspx (accessed 3 June 2022)
Monte Scuderi	Messina	n.a.	n.a.	n.a.	[47]
Bosco di Malabotta	Messina	Forest clearing	n.a.	C-P	S. Pasta, VI.2010
Fondachelli Fantina, Antillo e Francavilla di Sicilia	Messina	Forest clearing	n.a.	C-P	S. Pasta and A. Pipitone, VI.2003
PALERMO MTS.					
Versante settentrionale di Monte Fior dell’Occhio	Palermo	Streamside	*Ulmo canescentis–Salicetum pedicellatae*	SP	S. Pasta and A. Troia, VI.1989
Tratto superiore del Fiume Nocella	Palermo	Riverside	*Ulmo canescentis–Salicetum pedicellatae*	SP	S. Pasta, A. Troia and G. Pirrera, VI.1997

### 2.2. Traditional Medical–Veterinary Usages

As pointed out in [53], many medicinal uses reported in the main medical treatises dating back to Greek and Roman antiquity were transmitted by medieval monks and herbalists for centuries and became part of folk medicine even in areas with little cultural exchange. Elderberry did not escape this rule; for instance, the decoction of its leaves and flowers was used to treat ophthalmic infections, and poultices made from the leaves were long considered effective against erysipelas and other skin diseases both in Sicily and Sardinia. Moreover, between the 14th and 16th centuries, “Oleum sambucinum” was one of the twenty oils and distilled waters, mostly used for the treatment of skin diseases, that all the “aromatari” (=herbalists and apothecaries) of Palermo had in their store [54].

Between 1629 and 1815, *S. nigra* was used as a diaphoretic and diuretic by Sicilian apothecaries [55]. The authors of [56] reported plenty of additional usages of elderberry: “the decoction of its fronds solves cholera, and phlegm, its root cooked in wine helps those suffering from dropsy, it works against the bites of poisonous animals, the juice squeezed from the rind induces vomiting, the distilled water from the flowers mitigates the inflammation of the eyes, and if applied in small pieces it is useful on old ulcers; if the wood is harvested in the fifteenth of August and dried in the shade, then wearing a piece of it, like a cross hanging from the neck, prevents and heals any strong sore throat”.

An original manuscript full of folk remedies in use in Sicily during the 18th and 19th centuries and with many references to previous centuries [57] highlighted the extremely wide spectrum of local medicinal uses of elderberry, confirming that the root was part of the ingredients of a poultice used against dropsy, swollen legs and quartan fever; the young sprouts in decoction facilitated menstrual flow; the flowers were considered to be an effective remedy against stomach stones; a decoction of the inner part of bark was used as a laxative or applied against skin burn; and the leaves were pounded to make a preparation against mange.

Another author [58] enumerated the parts of *S. nigra* used for medicinal purposes (inner bark, leaves, flowers, preserve, water infusion, spirit, vinegar, berry extract, seeds and oil extract), their properties (narcotic, analgesic, purgative, sudoriferous, diuretic, promoting menstrual flow and galactagogue) and their applications against erysipelas (flower extract), diarrhoea (berries), dropsy, haemorrhoids (bark) and to prepare cataplasms.

It has been reported [59] that the sweat after taking a cup of elderberry tea was able to alleviate “the effects of a decillionth of a grain of arsenic”. The so-called “roob sambucino” was prepared from the elderberry fruits, whose juice was cooked until it acquired the consistency of a molasse. Once ready, this preserve was used “as sudorific, and in higher amounts it works as an excellent purgative; this action also is found in the bark” [60]. Celebrating the many potential usages of elderberry, the authors of [61] encouraged the “doctors of the poor and of the countrymen” to prescribe it, considering that each part of the plant causes a rather intense response in the target tissues and organs, e.g., causing vomiting or purging when applied to the gastrointestinal mucosa, stimulating diuresis, sweat, etc.

Another work [62] mentioned an exquisite wine obtained by mixing the fermented berries of *S. nigra* with sugar, ginger and cloves and stated that “the leaves, tender shoots and flowers purge vigorously. Infusion of the dried flowers promotes abundant skin transpiration. The leaves and flowers in poultice heal erysipelas, rheumatism and skin inflammation. The berries are diuretic and stop dysentery; the seeds have purgative properties and produce a resolving oil for external use”.

At the end of the 19th century, elderberry still played a major role in the Sicilian popular pharmacopeia. In fact, infusions of elderberry flowers were considered a very cheap and common remedy against colds (“arrifriddatura”) [63]. In addition, the compress obtained by adding starch to the green leaves of elderberry was defined as “sciugghienti lu sangu”, that is, capable of eliminating hematomas [63].

In Naso, the bark of elderberries was harvested at dawn, then pounded and left to infuse for 24 h. The so-called “elderberry water” obtained with this procedure was administered for three days as a remedy against childhood diarrhoea [63]. Moreover, “roob sambucino” was commonly sold by the apothecaries of Palermo [64].

Half a century later, the authors of [65] reported the amount of elderberry produced in several Sicilian municipalities, but unfortunately, they did not specify the identity of the harvested material nor the purposes, usages (Table 4) or costs of harvesting, transporting and processing. These data point out that elderberry still represented a meaningful economic resource for many rural communities until 70 years ago. The data reported in Table 4 show that elderberry was even abundant in areas for which the other sources consulted did not provide useful information about its occurrence. This result confirms the usefulness of conducting multidisciplinary research to build a comprehensive and correct picture of the distribution of the most common woody species, for which, paradoxically, the available knowledge is often fragmentary and imprecise.

**Table 4 plants-12-03457-t004:** Average annual elderberry production in Sicily during the 1950s; in bold: municipalities producing more than 1.5 tons per year (data from [65], modified).

Municipality	Province	Tons/Year
Cammarata	Agrigento	1.0
Casteltermini	Agrigento	1.0
Racalmuto	Agrigento	0.5
Raffadali	Agrigento	1.5
Santa Margherita Belice	Agrigento	0.3
Caltanissetta	Caltanissetta	0.5
Marianopoli	Caltanissetta	0.5
Mazzarino	Caltanissetta	2.0
Santa Caterina Villarmosa	Caltanissetta	0.5
Vallelunga Pratameno	Caltanissetta	0.5
Adrano	Catania	0.5
Bronte	Catania	0.5
Caltagirone	Catania	1.0
Randazzo	Catania	1.0
San Michele di Ganzaria	Catania	2.0
Zafferana Etnea	Catania	0.3
Enna	Enna	10.0
Leonforte	Enna	5.0
Nicosia	Enna	2.0
Piazza Armerina	Enna	5.0
Troina	Enna	0.5
Valguarnera Caropepe	Enna	1.0
Caronia	Messina	1.0
Castroreale	Messina	2.0
Cesarò	Messina	5.0
Floresta	Messina	1.0
Francavilla di Sicilia	Messina	2.0
Montalbano Elicona	Messina	5.0
Patti	Messina	2.0
Raccuja	Messina	1.0
Roccella Valdemone	Messina	0.5
Santa Lucia del Mela	Messina	0.1
San Piero Patti	Messina	1.0
Tortorici	Messina	2.0
Baucina	Palermo	0.5
Caccamo	Palermo	1.0
Castelbuono	Palermo	2.0
Cefalà Diana	Palermo	0.2
Lercara Friddi	Palermo	0.2
Marineo	Palermo	0.5
Mezzojuso	Palermo	0.2
Misilmeri	Palermo	0.3
Polizzi Generosa	Palermo	1.0
Prizzi	Palermo	1.0
San Cipirrello	Palermo	1.0
Scicli	Ragusa	0.5
Salemi	Trapani	1.5
Total		68.6

The ethnobotanical surveys carried out over the past 50 years allowed for the gathering of precious data, mostly resulting from interviews of elderly informants. Unfortunately, most of the recorded uses and the whole bulk of traditional knowledge have already disappeared or are disappearing almost everywhere in the Sicilian territory.

In the territory of Bivona [66], where a “fountain of the elderberry” (“funtana d’u savùcu”) still exists in the centre of the town (G. Clementi, pers. comm.), elderberry flowers were used in infusion as expectorants, while in the surroundings of Caltanissetta [67], the use of flower infusion as galactagogue and to fight influenza and bronchial diseases, gout and bloodshot eyes was reported, while the decoction of the bark was indicated against arthritis, as a depurative and diuretic, and the fruit decoction against constipation. In Cammarata, the same remedies were applied to people and domestic animals: swellings (“vintìi”) were combated with compresses of elderberry leaves and the same remedy was used to treat “u pilu â minna” (mastitis) in women who had just given birth [68]. Subjects suffering dermatosis manifesting as yellow and brown spots on the face, called inzavucati (but also “savucati” and “savuchigni”) [24], were treated by being given bread with elderberry flowers to consume [68]. Elderberries were also used for medicinal purposes in the territory of Sant’Angelo Muxaro [69].

In Mazara del Vallo, the leaves were used to treat sprains [70], while in Bronte, the extract of leaves or young sprouts macerated in vinegar for a few weeks was boiled, reduced to a thick paste and applied to relieve ostealgia by rubbing the affected part [71].

In the Hyblaean district, to fully benefit from their energic sudorific power, flowers would be gathered on St. John’s Day (so that the saint himself could bless them), or the day after the full moon of May, so that evil spirits could be driven away [72,73]. The flowers, infused, are used as a narcotic and galactagogue, while the juice of the fruits also acts as an antineuralgic [73].

At Mezzojuso, dried flowers were used to treat otitis and to remove ear plugs [74]; this use is reported for the whole Sicily in [26].

At Mussomeli, the flowers of *S. nigra* are used fresh or macerated in olive oil to treat—and to facilitate the healing of—wounds, haemorrhoids and burns, while leaves, either fresh or prepared in decoction, are applied to treat infant bruises and burns [75].

For the Trapani province, the authors of [76] mentioned the use of the leaves and flowers of elderberry macerated in vinegar as an antirheumatic, in which the berries were used as a purgative, and the authors of [77] reported the use of a bark and leaf decoction as an anti-inflammatory and healing agent, as well as for topical use, preparing compresses to be applied to bruises. The flower infusion was used to make compresses for eyes affected by erysipelas (“risibbula”), while the fumigation (“perfumi”) of flowers and leaves was used to clear the respiratory system. A decoction of the bark was used as a mouthwash and was considered an excellent odontalgic. Poultices with crushed leaves and buds, moistened with olive oil, were used against haemorrhoids. The product obtained from the bark and leaves placed to macerate for a month was used for rubs to relieve joint or rheumatic pain. Finally, the leaves pounded with salt and vinegar were used as a remedy against head lice.

As for the Madonie district, the authors of [78] generically mentioned the medicinal use of *S. nigra* in Collesano and San Mauro Castelverde, while the authors of [79] reported the use of flowers against erysipelas.

In the Nebrodi area (Alcara Li Fusi and Militello Rosmarino [80]; Tortorici: G. Galati Sardo, pers. comm.) the leaves of *S. nigra* has a soothing effect if rubbed on the irritations caused by nettles (*Urtica* spp.). At Cesarò, the fumigation with the leaves or the decoction used as a mouthwash is used as a sedative against toothache [81]. Referring to the medicinal and veterinary use of elderberry, the authors of [82] reported for Mistretta: “With the fresh or dried leaves boiled for about 20 min in water, a decoction is prepared to disinfect wounds. Boiled and rubbed on insect bites, they soothe itching; applied to abscesses they promote their rapid maturation; placed on bruised and swollen parts they reduce edema”, while the leaves are used as a diaphoretic to fight against conditions affecting the respiratory system. For the territory of the Nebrodi Park, the authors of [83] mentioned the use of a flower and fruit infusion as a diuretic.

The authors of [84] illustrated the wide range of applications of elderberry in the east-central part of Sicily: “infusion of the flowers is used against insomnia and erysipelas (Ragusa province), in influenza and febrile states (provinces of Ragusa, Catania, Enna and Caltanissetta) and to treat haemorrhoids (Ragusa and Enna). Decoction of the bark was used to fight infantile diarrhoea (Enna, Caltanissetta, Ragusa), to treat malaria (Ragusa), as a depurative and diuretic (Caltanissetta and Messina) and for external use against rheumatism (Messina). Fumigations of leaves and flowers were used to treat coughs (Caltanissetta)”.

For Monterosso Almo, the authors of [85] reported the past use of elderberry leaves, mixed with honey and olive oil, as a remedy against a neonatal pathology called “tassu”, whose main symptom was the stiffening of the palate nerve.

For the Sicani Mts., the authors of [86] reported the application of the decoctions and cataplasms of elderberry flowers and leaves to make poultices that were applied to treat hematomas, bruises, wounds, inflammations and swelling.

Finally, for the Etna area, the authors of [87] mentioned the use of an infusion of elderberry as a diuretic and provided a very long list of medicinal uses of *S. nigra*: as anti-inflammatory; as antipyretic against asthma, bronchitis, burns, colds and coughs; a depurative; a digestive; as diuretic; against eye inflammation, fever, gout pains, haemostatic, haemorrhoids, heart disorders and insect bites; as laxative, as lenitive; as migraine and rheumatic pain reliever; as sedative; and to treat skin and throat infections, toothache and wounds.

### 2.3. Traditional Food Uses

According to the authors of [88], regarding the elderberry featured in the “Herbarium cappuccinorum”, i.e., the garden of the Capuchin convent of Francavilla di Sicilia (province of Messina), its flowers were used raw by local friars to prepare omelettes and fritters, while their infusion was used to prepare a liqueur.

The fruits of *S. nigra* are eaten fresh or dried and used for the preparation of syrups, jams and juices; the fresh flowers are used to prepare fruit preserves, jellies, aromatic vinegars, are frozen and served as sorbet, or are fried in batter and covered with a sprinkling of sugar [78,89,90,91,92].

During the full blooming season of *S. nigra*, a festival dedicated to the “vastedda c’u sambucu” (=village bread with elderberries) takes place at Troina (province of Enna). Here, the raw elderberry flowers are kneaded to flavour local focaccia-like bread, stuffed with salami/pancetta and slightly soured fresh cow’s milk cheese [93,94,95]. The term “vastedda” describes a particular round bread with small rays, kneaded with elder flowers produced for Pentecost in Ragusa [96]. At Troina, it was originally prepared and consumed between late May and early June to celebrate St. Sylvester [97,98]. The same bread, named with different names such as “pani ê maju” (May bread) and “pani cu ’i çiuri” (=bread with flowers), is also produced and eaten [99] elsewhere—not only in the nearby surroundings—such as at Enna [25], Villarosa [100] and Piazza Armerina. In fact, elderberry flowers were used to flavour bread even in the villages of the Hyblaean district (people from Ragusa Ibla call it “scaccia c’o ciuri” (a sort of pizza with flowers) “https://blog.giallozafferano.it/fantasiaincucina/scaccia-con-il-fiore-di-sambuco-ricetta-siciliana/2 (accessed on 9 January 2023)” and in the Madonie Mts. [79]. Similar practices and products have also been recorded in several communities of inner western Sicily, such as at Mezzojuso [74] and Palazzo Adriano, where the bread is called “pani c’u savucu” “https://comunicalo.it/2023/05/13/savuco-experience-a-palazzo-adriano-un-tour-esperienziale-alla-scoperta-del-fiore-di-sambuco/) (accessed on 21 May 2023)”, and Castronovo di Sicilia, where it is called “cuddureddi c’u savucu” (G. Vacante, pers. comm.; https://www.siciliaagricoltura.it/2023/04/27/tutto-pronto-per-la-xvi-sagra-della-truscitedda-di-castronovo-di-sicilia/ (accessed on 2 May 2023)”. Moreover, the widespread use of the term “pastina” to define the elderberry inflorescence over a large portion of eastern Sicily (Table 1) confirms its common use as a food resource.

In another village of Enna province, Gagliano Castelferrato, elderberry flowers are used to produce a peculiar sparkling wine called “spumante di sambuco” (G. Galati Sardo, *pers. comm*.).

As for the Madonie territory, the authors of [101] also reported the use of elderberry seeds to decorate the surface of the “pan’i San Giuseppi”, a special bread produced to celebrate St. Joseph.

### 2.4. Notes on the Origin and the Meaning of the Terms “Sambuca” and “Zammù”

The similarity of the names makes many people believe that elderberry is one of the key ingredients of the famous liquor “Sambuca”. However, this belief appears to be completely unfounded [102]. In this regard, Luigi Manzi, the creator of sambuca, clarifies “I produce a fine anisette that does excellently to the stomach after a meal on account of the ‘sambuchelli’, the water vendors of his territory (i.e., Naples and Ischia Island, authors’ note) who go in the fields to quench the farmers’ thirst by bringing them water and aniseed” (Manzi family archival documents in [103]). The absence of any derivatives of *S. nigra* was confirmed to us by the staff of the Molinari Company, producer of the most famous brand of this liqueur. Research has still not confirmed the use of elderberry by Carthusian monks in Serra San Bruno (province of Vibo Valentia, Calabria) to make a liqueur [9]. In contrast, elderberry is among the ingredients listed on the labels of some liqueurs still on the market today, such as Opal Nera, produced at Novara in Piedmont [103], and Sambuca Romana. Moreover, the entries for “Sambuca” in the two most authoritative Italian dictionaries, i.e., [104] and Enciclopedia Treccani “http://www.treccani.it/vocabolario/sambuca2/ (accessed on 9 January 2023)”, read, respectively, “Liqueur, a kind of syrup and strong anisette” and “Liquor made from aniseed, flavoured with a distillate of elderberries and other essences, originally prepared by the ‘butteri’ (=cowboys) of the Campagna Romana (i.e., the plains of the Rome area), then produced industrially in various centres of Latium”. Indeed, at one time, the popular pharmacopoeia also included the so-called “elderberry wine (or roob)”, made by fermenting the fruits of *S. nigra*. This medicine, however, bore no similarity to anisette either in taste and smell or in colour, tending toward red, or in its curative properties. In fact, elderberry wine was not recommended as a digestive and carminative but as an emollient, diaphoretic and diuretic.

Even in the case of the traditional Sicilian anise liqueur called “zzammù” or “zambù” [25], the thesis that this product was originally obtained from the distillation of elder flowers and seeds appears to be completely unfounded [105].

### 2.5. The Multifold Uses of Elderberries in Rural Sicilian Everyday Life

At Naso [96], Alcara Li Fusi and Militello Rosmarino [80] and Tortorici (G. Galati Sardo, pers. comm.), elderberries were planted to demarcate property boundaries. This use is confirmed by the vernacular names “savucu di gai” and “savucu di sipala” (=hedge-making elderberry), also applied to *S. nigra* elsewhere in the Nebrodi district (see Table 1).

With reference to the quality of firewood, the authors of [72,106] reported the following proverb from Nicosia: “Se vói bien da to to mugghìe/Portighi ddigni d’fìghièra./Se ’a vói bien du ‘n tuttu/Pòrtighi ddigni di sambugu” (=if you love your wife, bring her fig wood; if you love her very much, bring her elderberry wood). According to G. Lo Papa, a forestry doctor from Nicosia, the local proverb points out that elderberry wood is ideal for baking because it burns slowly and has a high calorific value. Indeed, the authors of [107] confirmed that *S. nigra* provides excellent fuel wood. Moreover, elderberry wood is dense, heavy, hard, durable and exhibits a fine and very even texture. These characteristics give it excellent workability, which is why it ranks among turners’ woods; in particular, in the past, it was often used to produce cutlery [108].

Elderberry branches with split tips are used in the Nebrodi district as rudimentary tools to decorticate white and red willow suckers and other rods used for weaving work [80].

In Bronte, the fronds of elderberry and the “carducci” (the flowerheads of wild artichokes, authors’ note), are used to prepare a flavouring mixture called “cardunata”, used to eliminate the stench from empty wooden barrels” [109].

The generic name *Sambucus* is apparently derived from the ancient Greek term σαμβύκη (=sambúkē) or σαμβύχη (=sambúchē), which probably also gave rise to the Latin world “sambuca”, which defines either a small harp made with a bowed elderberry twig or a kind of flute made with the same material [110].

The soft central pith of young and straight elderberry branches can easily be drilled and stripped to make the insufflator of bagpipes, a purpose for which it was used extensively in the Nebrodi and Peloritani Mountains [111]. The protagonist of the novel “Jeli il pastore” [112] is a young shepherd who “plays an elderberry whistle”. The authors of [27] stated that because of its extreme lightness, elder wood was used for electrometers. In fact, the earliest electrometers, instruments capable of measuring the electrical charge possessed by a body, consisted of a ball made of elderberry pith hanging from a support by a silk thread [113]. The authors of [72] also reported the use of rubbing elderberry leaves on ropes to increase their durability in Sicily.

### 2.6. Elderberry Wood in Traditional Games and Toys

The authors of [80,109] reported that the boys of Bronte, Alcara Li Fusi and Militello Rosmarino used to build rudimentary compressed-air toy weapons, blowpipes called “cacciabballi” (or “caccia-baddi” [72]), using young branches (about 15 cm long and about 4 cm in diameter) of *S. nigra*. For this purpose, by means of a hot iron wire, they first pushed the pith (called “muddicuni” = crumb) out of the twig. Once a kind of hollow cane was obtained, projectiles (e.g., chickpeas or pieces of wet tow) were ejected by the pressure of pushing a 20 cm-long plunger made of much harder wood, usually hackberry (*Celtis australis* L.). The construction of similar toys is known throughout Sicily with a wide spectrum of vernacular names, such as “scupittuni” (=gun) [114,115,116] in Mezzojuso [72], “scattiòlu” (=“small banging object”) in Caltanissetta [94], “scassu” in Buccheri (same meaning, G. Garfì, pers. comm.) or just “savùcu” in the Madonie Mts. [79]. Similar toy-like weapons were built using elderberry wood almost everywhere in Italy, for instance in Brianza (northern Lombardy), where it is known by the name “schiupet”, or in Abruzzo and Molise, where it is called “scaricarelle”, “schizzo”, “schezzette”, “schiuppettuòle”, etc. http://www.giocopopolare.it/lo-schioppetto.html (accessed on 9 January 2023).

Other rudimental toy weapons may have used small pieces of elderberry as projectiles; this was the case for crossbows made with a stem of *Ferula communis* L. and a curved reed cane by the Albanian shepherds of Contessa Entellina and Piana degli Albanesi or that of the blowpipe used to shoot “badduzzi di ventu” (“small wind balls”) [115,116].

Elder wood tokens were used both as projectiles and as flags to signal the conquest of a hill disputed in turn by two teams of boys in Borgetto near Palermo [114].

The authors of [93] also described a peculiar toy built with a piece of elderberry wood called “misirrizziu” (from the Italian “mi si rizzi” = let me stand up). This rudimentary anthropomorphic puppet, well known across the Italian peninsula since at least the 17th century AD, often carried a small feather at the top. Being filled with lead, the lower hemispheric part of its body was strikingly heavier than the rest so that, when tilted, it promptly returned to an upright position.

### 2.7. Elderberry in Magical–Religious Practices and Beliefs

Phytotherapeutic practices and magic–religious beliefs cooccur and are tightly intermingled in the folklore of most Mediterranean countries, and the same happened in Sicily [63]. Not surprisingly, a medicinal plant used—and very effective—for an extremely wide spectrum of purposes, such as *S. nigra*, was also considered to have magic powers (see, for instance, [117]). As early as the 4th century BC, the authors of [118] accurately described the morphology of elderberry and the uses of some of its portions: for instance, the branches were used as walking sticks and as protection against snakes, while the juice extracted from the fruits “looks like wine, and men dip their hands and heads during the initiation to the mysteries”. In his translation of Dioscorides (1st century A.D.), the author of [119] reported that elderberry roots were also used to cure the bites of vipers, while the fresh leaves mixed with polenta were considered an effective remedy for dog bites. Similarly, the use of elderberry branches to kill snakes and/or to keep thieves away was reported [120]; until a few decades ago, it was reported that Sicilians still carried elderberry wood sticks with the same “double purpose” [121]. Today, however, any popular reminiscence of the magical and/or sacred role of elderberry appears to have completely faded across the whole island; although the frequent contiguity between elderberry stands and shrines is barely remarked upon, one hint of the past connection between religious rituals, sacred places, and elderberry has been traced. In fact, in the Colliri locality, near the village of Naso (Nebrodi Mts.), local devoted people used to carry elderberry sticks during the procession dedicated to the local patron saint, the Madonna delle Grazie, whose sanctuary, according to tradition, was built in a dense forest of elderberries [98].

Pitrè in [120] provides information about the use of elderberry during the celebration of Santa Rosalia in Palermo, when young boys tied leafy twigs of *S. nigra* around their heads and waists and rattles and bells on their temples. Until a few decades ago, elderberry twigs, rattles, feathers and colourful handkerchiefs still adorned the heads of the participants of the palio, a speed contest involving a large number of young boys in many Sicilian villages [93].

Moreover, there have been reports [63] of a peculiar popular belief from Montevago connected with the “magic aura” and the lifecycle of elderberry. Chloasma, also known as the “mask of pregnancy,” is a pigmentation disorder of the skin. When such brownish spots appeared on the face of a pregnant woman, she was defined “’nzavucata” (literally “elderberried”) because it was believed that she was under the influence of elderberry, so that the darker skin patches grew during the elderberry sprouting and blooming season, faded as its leaves were born and disappeared as they fell.

### 2.8. Between Traditional Use and Scientific Research on the Biochemical Properties of Elderberry: Confirmations and New Perspectives

Bibliographic and ethnobotanical surveys highlight the large use of different parts of an elderberry in Sicilian folk medicine. Numerous therapeutic properties have been proven by scientific studies conducted worldwide in recent decades. For example, biochemical and chemical studies on elderberry have shown the presence of numerous active ingredients [122,123,124]. On the other hand, the presence of toxic active ingredients in the bark and unripe fruits of black elderberry has been reported [125]. Elderberry leaves also contain a dangerous alkaloid, sambucin, as well as sambunigrin, a cyanogenic glycoside, used as a diaphoretic in diseases of the respiratory system. The flowers, used for the same purpose, contain volatile oils, saponins, glycosides, tannic and resinous compounds, mucilages, etc. [126].

The leaves contain many amino acids and antioxidants; the total amino acid content is 11.5%, where alanine, glutamic acid and aspartic acid dominate. The protein amount is higher in the leaves than in the flower extracts [127].

As of elderberry flowers, the authors of [128] mention their widespread use across theMediterranean countries as folk phytotherapeutic. The flowers exhibit antibacterial [129] and anti-inflammatory [130] activity, antiviral activity against influenza type A and B and herpes simplex [131] and diaphoretic and diuretic effects [132]. Moreover, the flower extracts of *S. nigra* contain water-soluble components that can stimulate insulin secretion and improve muscle glucose uptake and metabolism [133]. The infusions of elderberry flowers are known for their diaphoretic, anticatarrhal, expectorant and anti-inflammatory properties [134]. According to [133], the flowers contain flavonoids (up to 3%), mainly composed of flavonoid glycosides (astragalin, hyperoside, isoquercitrin and rutin); aglycone (quercetin and kaempferol); cyanogenic glycosides, including sambunigrin; triterpenes (about 1%), including α and β amyrin, oleanolic and ursolic acids; about 1% sterols (β-sitosterol, campesterol, stigmasterol); essential oils (0.15%), 66% of which are free fatty (linoleic and palmitic) acids; minerals (8–9%), mainly potassium; phenolic acids; chlorogenic (3%), caffeic and ferulic acids; tannins; mucilages; pectins; and sugar.

The chemical components of the fruits have received considerable interest; indeed, they have been used both as colouring agents for some foods and as agents against oxidative stresses [135]. They have a high anthocyanin content [134,136,137,138]. Numerous studies [139] have shown that these anthocyanins exhibit antioxidant activity and are potent inhibitors of lipid peroxidation. Some experiments on animals have shown the mild positive effects of the fruits on cholesterol and liver fatty acid levels [140,141,142]. The fruits also support the function of endothelial cells (thus preventing several vascular diseases) by protecting them against oxidative stresses [143]. Other studies showed that elderberry fruits may improve capillary network and brain function, lower plasma glucose levels, improve night vision and vascular retinopathy in diabetics and slow down the onset of lens opacities in patients with mild cortical cataracts [139]. The fruits apparently have a neuroprotective and antiaging effect too [144,145].

Elderberry bark is the first in which α-amyrenone was found in nature; it also contains α-amirin, betulin, oleanolic acid and β-sitosterol [146]. In addition, it contains SNA-I [147] and SNA-II lectins [148,149], which are ribosome-inactivating proteins (RIP) [150,151] that can also be used for plant protection acting as insecticides [152]. The bark also has antiviral activity [153], purgative and diuretic properties, and is used against epilepsy [20].

The flavonoids in the fruits exhibit anticancer, immunostimulant, antibacterial, antiallergic and antiviral properties [154]; their consumption can prevent the onset of cardiovascular disease, inflammation and diabetes [134,155]. The fruits contain trace amounts of essential oils, vitamins B6 and C, folic acid, β-carotene and pantothenic acid [145].

Standardised elderberry extracts obtained from elderberry fruits have demonstrated [156] influenza antiviral (in vitro) and anti-inflammatory properties through the production (in vivo) of anti-inflammatory and immune-stimulating cytokines (IL-1BETA, TNF-ALFA, IL-6 and IL-8). In another study carried out in vivo [134], a standardised elderberry extract reduced the duration of flu symptoms by 4 days.

Phytotherapeutic products containing elderberry have considerable popularity in both North America and Europe; for instance, OptiBerry IH141, (wild blueberry, strawberry, cranberry, wild blueberry, elderberry berry and raspberry extracts) is used as a supplement; Sinupret^®^ (containing *S. nigra* flowers, gentian root, verbena, primrose flowers and sorrel) is effective against both acute and chronic sinusitis; and Sambucol^®^ Active Defense (elderberry fruit extract, vitamin C, zinc, *Echinacea angustifolia*, *Echinacea purpurea* and propolis), Sambucol^®^ Immune System (elderberries, Echinacea angustifolia roots, Echinacea purpurea roots, propolis, vitamin C and zinc) and Sambucol^®^ for children (elderberries, *Echinacea purpurea* roots, *Echinacea angustifolia* roots, propolis) are used as antiviral and flu-fighting agents. In particular, some studies carried out on Sambucol^®^ have shown that it is able to stop the hemagglutination induced by influenza viruses [157]; it inhibits viral replication and protects against oxidative stresses. 

Recently, during the respiratory syndrome SARS-CoV-2 (COVID-19) pandemic, the interest in elderberry supplements is increased [158] for its potential use in alternative and complementary therapies to support the prevention and/or treatment of the upper respiratory symptoms, complications and adverse events caused by this severe illness.

## 3. Materials and Methods

Targeted bibliographic searches [24,25] were carried out to identify the main vernacular names referring to *S. nigra*. Toponyms provide botanists very valuable yet underrated information about landscape history and the use of its resources [159,160,161,162]. Hence, once the main vernacular names linked with the target species were identified, a specific query on the IGMI (Istituto Geografico Militare Italiano) database allowed for the identification and location of the toponyms associated with the regional occurrence of elderberry.

To outline the current distribution of elderberry in Sicily, unpublished records were added to occurrence data available from the regional floristic and vegetation scientific literature.

Information about the traditional usages of the elderberry in Sicily was obtained by consulting the ethnobotanical, ethnoanthropological and botanical literature. Data deriving from this research are reported following the chronological order of the sources. Supplementary information about the past and present usages of *S. nigra* is the result of several interviews; a complete list of the informants is provided in the acknowledgments.

An extensive online literature search focused on scientific papers published during recent years was performed by screening Google Scholar, Scopus and ISI Web of Knowledge using the keywords: *“Sambucus nigra*” plus “traditional medicine” plus “phytochemistry”.

## 4. Conclusions

Considering the lack of archaeobotanical evidence referring to elderberry, this species should better be interpreted as a useful plant that was introduced to Sicily from other European regions during the Neolithic period or in even more recent times. In more recent times, its expansion may have been facilitated by birds’ dispersal, as in the case of other trees whose native status remains doubtful, such as *Celtis australis* L., *Laurus nobilis* L. and *Viburnum tinus* L. [163].

While elderberry was widely used throughout the island in the past, its local uses currently appear sparse and sporadic, being limited to the food use of its flowers (to flavour bread or fry in batter), while there are no recent reports on the use of other parts of the plant for medicinal or dietary purposes. The gradual disappearance of local traditional knowledge clashes with the results of a recent study on the active ingredients present in the Sicilian fruits of *S. nigra*, which are particularly rich in anthocyanins with remarkable antioxidant capacities (A. Maggio, unpubl. data). However, despite the numerous properties of all parts of elderberry, to date its use is becoming rarer and rarer, with a very low and localised number of applications. Moreover, thanks to multidisciplinary research, the data retrieved from different sources allowed the current and potential distribution of the study species to be better traced. Such research also highlights the great potential of this plant and the urgent need to pursue interdisciplinary studies on the phytochemical and biological activity of its different parts.

## Figures and Tables

**Figure 1 plants-12-03457-f001:**
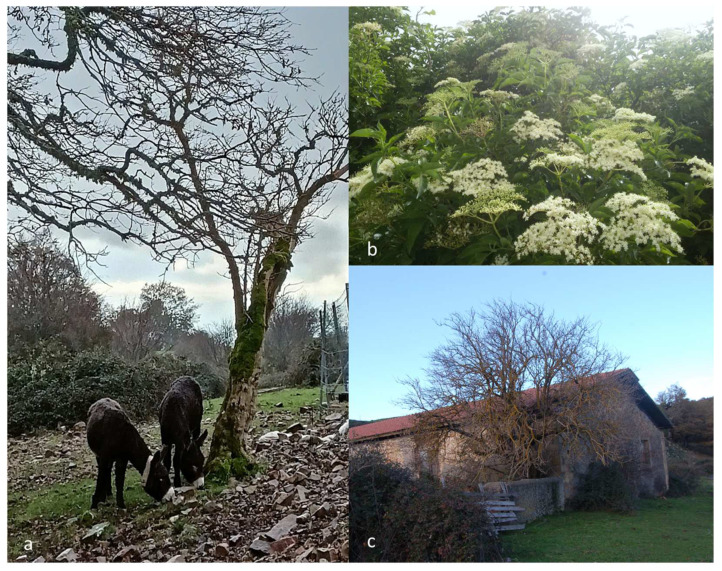
(**a**): Elderberries represent a common feature of traditional agro-pastoral landscapes; (**b**) pioneer stands of Sambucus nigra often dominate in hedges and forest fringes; (**c**) isolated individuals of elderberries used to be planted near rural houses.

**Figure 2 plants-12-03457-f002:**
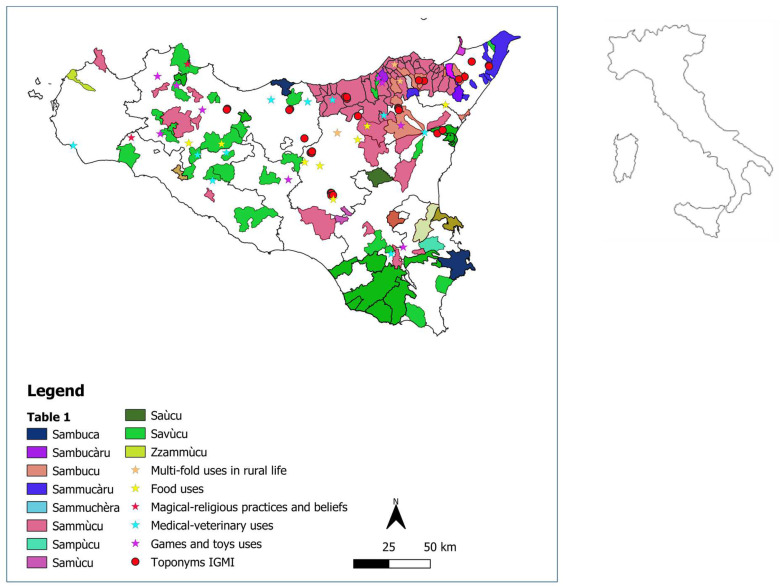
Distribution of the Sicilian vernacular names used to refer to elderberry and traditional use.

**Table 1 plants-12-03457-t001:** List and distribution of Sicilian vernacular names directly or indirectly used to refer to *Sambucus nigra* [24,25].

Phytonym	Meaning and Distribution
Sambuca	*S. nigra*: Cassibile. A wooden toy weapon: Cefalù
Sambucàru	*S. nigra*: Antillo, San Marco di Milazzo, Rodì-Milici
Sambùcu	*S. nigra*: Cefalù, Taormina, Milazzo, Rodì-Milici, Castroreale, Mazzarrà Sant’Andrea, San Piero Patti, Raccuja, Sinagra, Ucria, Galati Mamertino, Bronte, Carlentini, Cassibile
Sammucàru	*S. nigra*: eastern dialects of the Ionian and Tyrrhenian sector of Messina province: Itala, Alì, Limina, Rometta, Mandanici, Messina (Faro Superiore), Floresta, Antillo, Rodì-Milici, San Marco di Milazzo.
Sammuchèra	*S. nigra*: Ucria
Sammùcu	*S. nigra*: central and western dialects of Messina province: Fondachelli-Fantina, Novara di Sicilia, Furnari, Mazzarrà di S. Andrea, Tripi, Falcone, Oliveri, Basicò, Montalbano Elicona, Patti, Librizzi, S. Piero Patti, Montagnareale, Gioiosa Marea, Piraino, S. Angelo di Brolo, Raccuja, Brolo, Ficarra, Sinagra, Ucrìa, Floresta, Tortorici, Castell’Umberto, Naso, Capo d’Orlando, Capri Leone, Mirto, Frazzanò, S. Salvatore di Fitàlia, Galati Mametino, Longi, S. Marco d’Alunzio, Torrenova, Militello Rosmarino, Alcara li Fusi, S. Agata di Militello, Acquedolci, Cesarò, S. Teodoro, S. Fratello, Capizzi, Caronia, S. Stefano di Camastra, Reitano, Mistretta, Castel di Lucio, Motta d’Affermo, Pettineo, Tusa; Linguaglossa, Adrano, Paternò, Militello Val di Catania, Grammichele, Augusta, Cassaro, Giarratana, Troina, Regalbuto, Mazzarino, Marineo, Corleone, San Giuseppe Jato, Raffadali, Calamonaci, San Vito Lo Capo
Sampùcu	*S. nigra*: Carlentini, Sortino
Samùcu	*S. nigra*: Mirabella Imbàccari, San Michele di Ganzarìa
Saùcu	*S. nigra*: Mascali, Sant’Alfio, Riposto, Giarre, Sferro (Giarre), Castel di Judica, Militello Val di Catania, Augusta, Palazzolo Acreide, Canicattini Bagni, Chiaramonte Gulfi, Acate, Ragusa, Modica, Scicli, Santa Croce Camerina, Altofonte, Ustica. Impetigo: Montemaggiore Belsito.
Savùcu	*S. nigra*: Licodia Eubea, Acate, Modica, Ispica. A wooden toy weapon: Avola, Palermo. Freckles: Santa Caterina Villarmosa, San Cataldo, Alia, Ventimiglia di Sicilia, Baucina, Piana degli Albanesi, Campofiorito, Castronovo di Sicilia, Roccamena, Bisacquino, Campofiorito, Naro, Bivona, San Giovanni Gemini, Casteltermini, San Biagio Platani, Villafranca Sicula, Calamonaci, Menfi. “*Pastina*” (=elderberry inflorescence): Mascali, Sant’Alfio, Riposto, Giarre, Ragalna, Castel di Judica, Militello Val di Catania, Augusta, Palazzolo Acreide, Canicattini Bagni, Chiaramonte Gulfi, Acate, Ragusa, Modica, Scicli, Santa Croce Camerina, Montemaggiore Belsito, Altofonte, Ustica
Zzammùcu	A ballad: Erice

**Table 2 plants-12-03457-t002:** Distribution and environmental characteristics of the Sicilian place names that refer to *Sambucus nigra* found on the official maps of IGMI.

Place Name	Type of Site	Municipality	Province
Savuca (Abbeveratoio)	Drinking trough	Canicattì	Agrigento
Savuco (Sorgente)	Freshwater spring	Caltanissetta	Caltanissetta
Sambuco (Punta, Casa, Vallone)	Mountain top + isolated house + watershed	Maniace	Catania
Sambuco (Casa)	Isolated house	Mascali	Catania
Sambuco (Torrente)	Stream	Milo	Catania
Saugo (Abbeveratoio)	Drinking trough	Vizzini	Catania
Sambuco (Case, Casazze, Contrada, Vallone)	Isolated houses + locality + watershed	Calascibetta	Enna
Sambuco (Fosso)	Watershed	Cerami	Enna
Sambuco (Contrada, Màrcato, Monte)	Locality + isolated houses + mountain top	Piazza Armerina	Enna
Saucco (Case, Monte)	Isolated houses + mountain top	Alì	Messina
Sambuco (Contrada, Cozzo, Casa)	Locality + mountain top+ isolated house	Caronia	Messina
Sambuca (Contrada, Pizzo)	Locality + mountain top	Castroreale	Messina
Sambuco (Valle)	Watershed	Mela	Messina
Sambuco (Pizzo, Puntale)	Mountain top	Monforte San Giorgio	Messina
Sambuco (Monte)	Mountain top	Montalbano Elicona	Messina
Sambucazzo (Contrada, Vallone, Casa)	Locality + watershed + isolated house	San Fratello	Messina
Sambuco (Contrada, Monte)	Locality + mountain top	San Piero Patti	Messina
Sambuco (Valle)	Watershed	Santa Lucia del Mela	Messina
Savoca (Torrente)	Stream	Savoca	Messina
Sambuco (Contrada)	Locality	Blufi	Palermo
Sambuchi (Contrada, Serra)	Locality + ridge	Caccamo	Palermo
Sambuco (Case)	Isolated houses	Petralia Soprana	Palermo
Sambuchi (Case, Vallone)	Isolated houses + watershed	Petralia Sottana	Palermo

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
