# Peer review of "Sambucus nigra L. (fam. Viburnaceae) in Sicily: Distribution, Ecology, Traditional Use and Therapeutic Properties"

_plants, 2023, doi:10.3390/plants12193457_

Round 1

Reviewer 1 Report

In this manuscript (plants-2566798) entitled ‘Sambucus nigra L. (fam. Adoxaceae) in Sicily: distribution, ecology, traditional use and therapeutic properties’ submitted to Plants, Giovanna Sala and colleagues have examined the uses of elderberries and the beliefs surrounding them in the biomedical and pharmacological fields. This topic is interesting, and I have some minor concerns to be addressed to improve the quality of this manuscript.

1. Pictures to show the growth of elderberries in Sicily (Italy) should be provided in the revised manuscript.

2. In the table 4, authors showed the average annual Elderberry production in Sicily during the 1950s. These outdated data should be updated, what is the situation for the recent years?

3. English and wording should be profoundly improved. In some cases, e.g. line 330, ...[79] also report the use of elderberry seeds to decorate... it is unformal for a manuscript to be published.

4. Perspectives for improving uses of elderberries in Sicily should be provided in the revision.

Moderate editing of English language required

Reviewer 2 Report

> Thank you. I have been reading the manuscript. I support publication.> I would like to point out that the title should be changed. The> correct name for family Adoxaceae is now family Viburnaceae.>> The manuscript is well written and organized. The amount of detail is> amazing. There are a few places where words seem to be missing. A> close reading by an L1 speaker of English could clear this up.

There are a few places where words seem to be missing. A> close reading by an L1 speaker of English could clear this up.

Reviewer 3 Report

This review article on the traditional uses and pharmacological activities of Sambucus nigra (elderberry) in Sicily seems to be quite comprehensive and valuable in terms of shedding light on the historical uses of this plant and integrating traditional knowledge with contemporary scientific findings. Here are some comments and observations about different aspects of the article.

Abstract should show some main findings and concise methods of the study.

While the article discusses the traditional uses and pharmacological activities of Sambucus nigra, it would benefit from a clearer articulation of its specific objectives and scope. What exactly is the primary focus of the study? Is it to bridge traditional knowledge with modern science, explore the historical context, or present a comprehensive overview of elderberry's applications? Clarifying this can provide readers with a more precise understanding of what to expect.

The above points should be address in abstract and introduction.

Line 45 could be cited with some recent studies https://doi.org/10.3390/polym14142899, https://doi.org/10.1016/j.heliyon.2023.e15909

The article mentions that elderberry's pharmacological activities have been the focus of active research, but it doesn't delve into the specific findings or outcomes of this research. Providing more detailed information about the specific compounds, mechanisms of action, and the outcomes of relevant studies would enhance the scientific depth of the article.

While it's noted that elderberry has been used for treating numerous diseases, the article lacks specific examples of these traditional uses. Incorporating anecdotes, historical cases, or even interviews with traditional practitioners could bring life to the historical context and provide more substance to the claims being made.

The article mentions that the study aims to examine the traditional uses of elderberries in light of recent scientific findings. However, the article could strengthen this link by directly connecting traditional uses with relevant modern research findings. This would demonstrate the continuity of usage through time and provide credibility to both traditional knowledge and scientific research.

Line 57-61 are not clear may be need language improvement.

Most of the times species names are not italic which need to be italic.

“The map of Figure 1 shows the spatial” replace of with in

Line 51 should be cited with recent studies such as

http://doi.org/10.36899/JAPS.2022.3.0484, https://doi.org/10.1016/j.scitotenv.2022.160930

The article discusses the distribution of S. nigra in Sicily, but it could provide more comprehensive information, such as altitude ranges, soil preferences, and climate conditions where the plant thrives. These details can help readers better understand the ecological context of the plant's growth.

While the article mentions multidisciplinary research, it doesn't clearly outline how this data was integrated or what specific insights were gained from this approach. Explaining how the various disciplines contributed to the overall understanding of elderberry's uses and distribution would add depth to the discussion.

Line 57-61 are not clear may be need language improvement.

Most of the times species names are not italic which need to be italic.

“The map of Figure 1 shows the spatial” replace of with in

Round 2

Reviewer 3 Report

The authors addressed all comments successfully and the paper can be accepted. Italicize plant names or species names in the text and in the references as well.